# *Limosilactobacillus reuteri* Attenuates Atopic Dermatitis via Changes in Gut Bacteria and Indole Derivatives from Tryptophan Metabolism

**DOI:** 10.3390/ijms23147735

**Published:** 2022-07-13

**Authors:** Zhifeng Fang, Tong Pan, Hongchao Wang, Jinlin Zhu, Hao Zhang, Jianxin Zhao, Wei Chen, Wenwei Lu

**Affiliations:** 1State Key Laboratory of Food Science and Technology, Jiangnan University, Wuxi 214122, China; zhifengf@foxmail.com (Z.F.); 6200113202@stu.jiangnan.edu.cn (T.P.); hcwang@jiangnan.edu.cn (H.W.); wx_zjl@jiangnan.edu.cn (J.Z.); zhanghao61@jiangnan.edu.cn (H.Z.); zhaojianxin@jiangnan.edu.cn (J.Z.); chenwei66@jiangnan.edu.cn (W.C.); 2School of Food Science and Technology, Jiangnan University, Wuxi 214122, China; 3National Engineering Research Center for Functional Food, Jiangnan University, Wuxi 214122, China; 4Wuxi Translational Medicine Research Center and Jiangsu Translational Medicine Research Institute Wuxi Branch, Wuxi 214122, China; 5(Yangzhou) Institute of Food Biotechnology, Jiangnan University, Yangzhou 225004, China

**Keywords:** *Limosilactobacillus reuteri*, tryptophan metabolism, indole derivatives, aryl hydrocarbon receptor, gut microbiota

## Abstract

Gut bacteria are closely associated with the development of atopic dermatitis (AD) due to their immunoregulatory function. Indole derivatives, produced by gut bacteria metabolizing tryptophan, are ligands to activate the aryl hydrocarbon receptor (AHR), which plays a critical role in attenuating AD symptoms. *Limosilactobacillus reuteri*, a producer of indole derivatives, regulates mucosal immunity via activating the AHR signaling pathway. However, the effective substance and mechanism of *L. reuteri* in the amelioration of AD remain to be elucidated. In this research, we found that *L. reuteri* DYNDL22M62 significantly improved AD-like symptoms in mice by suppressing IgE levels and the expressions of thymic stromal lymphopoietin (TSLP), IL-4, and IL-5. *L. reuteri* DYNDL22M62 induced an increase in the production of indole lactic acid (ILA) and indole propionic acid (IPA) via targeted tryptophan metabolic analysis and the expression of AHR in mice. Furthermore, *L. reuteri* DYNDL22M62 increased the proportions of *Romboutsia* and *Ruminococcaceae* NK4A214 group, which were positively related to ILA, but decreased *Dubosiella*, which was negatively related to IPA. Collectively, *L. reuteri* DYNDL22M62 with the role of modulating gut bacteria and the production of indole derivatives may attenuate AD via activating AHR in mice.

## 1. Introduction

Atopic dermatitis (AD) is a common inflammatory disease in the skin and affects many infants, children, and even adults worldwide [1]. Susceptibility to AD is closely associated with genetic and environmental factors that increase the dysfunction of the epidermal barrier and/or dysregulation of the immune response, and mutations in the filaggrin gene are the strongest genetic predisposing factors to induce dysfunction of the epidermal barrier in AD [2,3]. Furthermore, there are massive amounts of bacteria in the skin and gut, and they affect and regulate the immune and healthy states of the host. Skin microbiome such as *Staphylococcus aureus* disturbs the epidermal barrier and induces decreases in antimicrobial peptides, an increase in T helper 2 (Th2) type cytokines, and disturbance of skin lipid metabolism [4,5]. In addition, the intestine is another important bacterial habitat, and gut microbial alteration is closely associated with allergic diseases including asthma, food allergy, and AD [6,7]. Gut microbial dysbiosis is an important cause of the onset of AD and this has been demonstrated in clinical studies [8,9]. *Clostridium difficile*, *Clostridia*, *S. aureus*, *Escherichia coli*, and obligate anaerobe species were prevalent but *Lactobacillus*, *Bifidobacterium*, *Akkermansia muciniphila*, and *Ruminococcus gnavus* were reduced in the feces of patients with AD [10,11]. It has been reported that the proportion of *E. coli* was positively correlated to serum immunoglobulin (Ig) E levels [12], but *Bacteroides fragilis* decreased IL-4 levels produced by CD4+ T cells in germ-free mice [13]. In a cohort of 24 infants, compared to healthy controls, the proportion of bacilli was significantly higher in AD infants. Furthermore, *Clostridia*, but not bacilli and *E. coli*, is significantly related to age at AD onset and negatively correlated with the proportion of eosinophils in the blood [14]. However, compared with patients with AD (aged 6–22 years old), the relative abundance of *Clostridium* was higher, but *Blautia* and *Parabacteroides* were lower in healthy controls [15]. There are significantly different gut microbial taxa in infants with AD compared to that in child/adult patients. Therefore, these specific changes influence gut microbial metabolisms and thus affect the balance between T helper 1 (Th1)- and Th2-type as well as Th17 and T regulatory (Treg) immune responses in AD [16].

Tryptophan metabolism involves the development of many diseases. Changes in tryptophan metabolites significantly affect the clinical characteristics of patients with food allergies [17], depression [18], and cancer [19]. Fructooligosaccharides significantly increased kynurenine levels and restored Th17/Treg balance to alleviate the clinical manifestations in mice with ovalbumin-induced food allergies [20]. Fuzhuan brick tea polysaccharide increased indoleacetic acid (IAA) and indole-3-aldehyde (IAld) levels and thus activated the aryl hydrocarbon receptor (AHR) signaling pathway to improve ulcerative colitis via producing interleukin-22 (IL-22) and evaluating the expression of intestinal tight junction proteins [21]. Gut microbiota converts tryptophan into indole and indole derivatives, which exert various physiological features including anti-allergy [22,23,24,25]. In germ-free mice with AD, reduced AHR signaling expression induced an increase in epidermal barrier dysfunction, which was restored using AHR agonist treatment [26]. Indole derivatives such as IAld, as an endogenous ligand for AHR, have been demonstrated to activate AHR signaling to alleviate psoriasis and AD [27,28]. Tryptophan metabolism is reduced in patients with AD, and treatment with IAld-activating AHR to inhibit aberrant Th2-type response improves AD-like symptoms [29,30]. Therefore, targeting activating AHR is an alternative way to ameliorate AD-like clinical symptoms. *Limosilactobacillus reuteri*, a tryptophan-metabolizing bacterium, produced indole derivatives and protected against *Candida albicans* colonization and intestinal mucosal inflammation via activating the AHR-IL-22 axis [31]. However, increased IL-22 is positively related to the development of AD via modulating the gene expression of the skin barrier-related molecules including filaggrin, loricrin, and involucrin [32], and its main producers are Th22 cells that do not produce IL-17A [33]. IL-22 production from Th22 and innate lymphoid cells (ILC) such as ILC22 is dependent on AHR [34,35]. AHR-IL-22 axis signaling pathway may not be the main mechanism to alleviate AD and thus the mechanism of action needs to be elucidated. We assumed that *L. reuteri* strains could alleviate the clinical symptoms via AHR activation. Furthermore, the effects of gut microbiota should not be excluded from the process. Therefore, combined with these alterations, this study aimed to explore: (1) the effects of *L. reuteri* strains on AD symptoms and (2) the mechanism of *L. reuteri* strains in alleviating AD via the gut microbiota.

## 2. Results

### 2.1. L. reuteri Strains Affected the Pathological Symptoms in Mice

To evaluate the effects of *L. reuteri* strains on the pathological symptoms of AD in vivo, the model was constructed using 2,4-dinitrofluorobenzene (DNFB) solution treatment. The experimental design is shown in Figure 1A. DNFB treatment significantly induced ear swelling in mice compared to the control group (Figure 1B). These pathological features were recovered by *L. reuteri* DYNDL22M62 and FSDLZ12M1 treatments versus the DNFB group. However, *L. reuteri* GDLZ105 and FWXBH12M3 treatments could not significantly reduce ear thickness. Furthermore, the skin section was stained using hematoxylin and eosin (H&E) solution to evaluate pathological alterations of skin in AD-like mice. Compared to the control group, DNFB treatment significantly aggravated the inflammatory infiltration and induced an increase in skin thickness, and *L. reuteri* strain treatments alleviated inflammation of the skin except for *L. reuteri* GDLZ105 (Figure 1C). These results showed that *L. reuteri* strain treatments, particularly, *L. reuteri* DYNDL22M62, significantly contributed to the alleviation of the pathological manifestations in AD-like mice.

### 2.2. L. reuteri DYNDL22M62 Suppressed Aberrant Immune Response

To evaluate the immunoregulatory effects of *L. reuteri* treatments on mice, IgE levels and Th2-type immune indicators were measured. In the DNFB group, IgE levels were significantly increased versus the control group (Figure 2A). *L. reuteri* strains significantly reduced IgE levels versus the DNFB group except for *L. reuteri* GDLZ105. Furthermore, DNFB treatment significantly elevated the expression of TSLP in the skin tissue versus the control group, and *L. reuteri* DYNDL22M62 treatment significantly suppressed TSLP levels versus DNFB treatment in AD-like mice (Figure 2B). *L. reuteri* DYNDL22M62 and FSDLZ12M1 significantly reduced IL-4 (Th2-type cytokine) compared with DNFB treatment (Figure 2C), and *L. reuteri* GDLZ105 and FWXBH12M3 could not suppress IL-4 levels. Although *L. reuteri* DYNDL22M62 (*p* = 0.17), FSDLZ12M1, and FWXBH12M3 reduced the expression of IL-5 compared to the DNFB group, there was no statistical significance (Figure 2D). Conversely, *L. reuteri* GDLZ105 significantly increased IL-5 levels versus the DNFB group (*p* < 0.05). Therefore, *L. reuteri* DYNDL22M62 reduced serum IgE levels and reduced Th2 type response in AD-like mice.

### 2.3. L. reuteri DYNDL22M62 Increased Production of ILA and IPA and AHR Expression

To explore the effective substance of *L. reuteri* treatments to alleviate AD-like symptoms, targeted tryptophan metabolism analysis was performed using UHPLC Q-Exactive-MS determination, DNFB treatment altered tryptophan metabolites in fecal samples compared to the control group, and indole derivatives including indolelactic acid (ILA), IAld, IAA, indoleacrylic acid (IA), and indole propionic acid (IPA) levels were reduced in the DNFB group (Figure 3). *L. reuteri* DYNDL22M62 treatment significantly increased ILA and IPA levels in fecal samples versus the DNFB group (Figure 3A,E), and IAld, IAA (*p* = 0.18), and IA (*p* = 0.20) showed the increasing trend after *L. reuteri* DYNDL22M62 treatment (Figure 3B–D). All other *L. reuteri* strain treatments contributed to the increasing trend in IAA and IPA but had no significant effects on ILA, IAld, and IA compared to the DNFB group. ILA and IPA were endogenous ligands for AHR, and then the expression of AHR was assessed in mice. The results showed that DNFB treatment significantly reduced the expression of AHR compared to the control group, but it was restored by *L. reuteri* DYNDL22M62 treatment (*p* < 0.05) (Figure 3F). *L. reuteri* GDLZ105 treatment significantly decreased the expression of AHR versus the DNFB group, and this was consistent with the severity of the AD-like symptoms. The expression of AHR decreased in the other two strain-treated groups although there was no statistical significance. These results implied that *L. reuteri* DYNDL22M62 might activate AHR via increasing the levels of endogenous ligands including ILA and IPA for AHR from tryptophan metabolism in AD-like mice.

### 2.4. L. reuteri DYNDL22M62 Treatment Reshaped the Gut Microbial Composition

*L. reuteri* DYNDL22M62 significantly altered gut microbial metabolism according to metabonomics analysis, and this was closely associated with changes in gut microbiota. To analyze the alterations of gut microbiota in the control, DNFB, and *L. reuteri* DYNDL22M62 groups, a high throughput sequencing for gut microbiota was performed. The indicators of alpha diversity were no significant difference between groups (Appendix A). Principal component analysis (PCA) showed the differences between samples in the control, DNFB, and DYNDL22M62 groups (Figure 4A). Furthermore, DNFB decreased the proportion of bacteria at the phylum level compared to the control group, and *L. reuteri* DYNDL22M62 treatment restored gut microbial abundance (Figure 4B). DNFB treatment reduced the abundances of *Actinobacteria*, *Firmicutes*, and *Verrucomicrobia* in AD-like mice, and supplement with *L. reuteri* DYNDL22M62 significantly increased the abundance of *Actinobacteria* (Figure 4C). At the family level, DNFB led to a decrease in the abundance of *Lactobacillaceae*, *Erysipelotrichaceae*, *Akkermansiaceae*, and *Bifidobacteriaceae* versus the control group. The abundances of *Erysipelotrichaceae* and *Peptostreptococcaceae* were significantly increased in the *L. reuteri* DYNDL22M62 group. Corresponding to the change in *Verrucomicrobia*, the abundance of *Akkermansiaceae* was increased by *L. reuteri* DYNDL22M62, although the difference was not significant between the DNFB and DYNDL22M62 groups. Additionally, there was an increasing trend in the abundance of *Lactobacillaceae* (*p* = 0.18) and *Bifidobacteriaceae* after *L. reuteri* DYNDL22M62 treatment. These results showed that *L. reuteri* DYNDL22M62 treatment regulated gut microbial composition in mice, and this might induce changes in tryptophan metabolism in fecal samples.

### 2.5. L. reuteri DYNDL22M62 Treatment Regulated the Differential Taxa Related to Indole Derivative Metabolism

Linear discriminant analysis effect size (LEfSe) was carried out to find the differential taxa among the control, DNFB, and DYNDL22M62 groups. Compared to the control group, *Prevotellaceae*UCG_001, *Alistipes*, and *Rikenellaceae* were enriched in the DNFB group, but *Bifidobacteriaceae*, *Dubosiella*, and *Bifidobacterium* were significantly decreased (Figure 5A,B). However, *Romboutsia*, *Ruminococcaceae* NK4A214 group, and *Peptostreptococcaceae* were the differential taxa in the DYNDL22M62 group versus the other two groups. *L. reuteri* DYNDL22M62 treatment significantly increased the abundances of the *Ruminococcaceae* NK4A214 group and *Romboutsia* but reduced *Dubosiella* versus the DNFB group (Figure 5B). In Figure 5C, ILA was positively related to the differential taxa including *Romboutsia*, *Bifidobacterium*, *Ruminococcaceae* NK4A214 group, and *Dubosiella*; and IPA was positively correlated with the former three gut bacteria but was negatively correlated with *Dubosiella*. Furthermore, IA was negatively related to Romboutsia and *Ruminococcaceae* NK4A214 group and positively related to *Bifidobacterium* and *Dubosiella* (*p* > 0.05). IAA was positively related to *Romboutsia* and *Bifidobacterium* but negatively related to the *Ruminococcaceae* NK4A214 group and *Dubosiella*. IAld was positively related to all the differential taxa but the correlations with the *Ruminococcaceae* NK4A214 group and Bifidobacterium were not significant (*p* > 0.05). These results implied that *L. reuteri* DYNDL22M62 treatment regulated tryptophan metabolism by affecting the specific differential taxa in AD-like mice.

## 3. Discussion

This study explored the effects of *L. reuteri* strains with the role of regulating gut microbial tryptophan metabolism on AD-like symptoms and the underlying mechanism in mice. In AD-like mice, a decrease in the integrity of the skin barrier and inflammatory infiltration due to aberrant immune responses. Therefore, AD was characterized by skin inflammation and excessive Th2-type responses in the acute stage [36]. *L. reuteri* strain has the potential to alleviate AD symptoms via modulating immune responses. *L. reuteri* Fn041, a strain secreting IgA, significantly improved skin swelling and inflammatory cell infiltration via restoring the systemic Th1 and Th2 cytokine ratios and Treg proliferation [37]. Likewise, *L. reuteri* DYNDL22M62 variably affected AD-like symptoms, which could be shown based on H&E staining and changes in cytokines. Treatment with *L. reuteri* DYNDL22M62 significantly reduced ear thickness, skin swelling, and inflammatory infiltration in skin lesions suggesting that *L. reuteri* DYNDL22M62 might have the function of regulating immunity (Figure 1B,C). *L. reuteri* strains significantly suppressed IgE levels except for GDLZ105, and *L. reuteri* DYNDL22M62 reduced Th2 cytokines including TSLP, IL-4, and IL-5 in AD-like mice (Figure 2). IgE-mediated hypersensitivity was one of the mechanisms to induce allergic diseases including AD [38]. IL-4 binds to IL-4Rα, which is expressed in T cells, B cells, and macrophages leading to excessive Th2 cell differentiation and IgE class switching [39]. DNFB-induced an increase in IL-4 led to a higher level of IgE and Th2 responses (Figure 2). Treatments with *L. reuteri* strains except for GDLZ105 reduced skin swelling and inflammation (Figure 1C), and this was consistent with the alteration of Th2 cytokines. TSLP, expressed in epithelial cells, is closely associated with the initiation and maintenance of AD and epidermal barrier integrity [40,41]. *L. reuteri* DYNDL22M62-induced a reduction of TSLP was conducive to improving skin lesions in AD-like mice. Collectively, these results indicated that *L. reuteri* DYNDL22M62 alleviated AD-like symptoms via restoring aberrant immune responses.

Indole derivatives are the main metabolites of tryptophan in the gut due to the catabolism of anaerobic bacteria and are associated with the maturation and function of the immune system. *L. reuteri* strains WU and 100-23 induced intraepithelial T lymphocytes via the release of AHR ligands including ILA and reprogrammed intraepithelial CD4 T cells into immunoregulatory T cells, suggesting the distinct AHR-mediated immunoregulatory mechanism of indole derivatives [42]. Treatments with *L. reuteri* DYNDL22M62 increased the tryptophan metabolism of gut microbiota and significantly induced the release of ILA and IPA (Figure 3A–E). Furthermore, *L reuteri* DYNDL22M62 increased the 1.35-fold-change expression of AHR compared to the DNFB group (Figure 3F). The AHR pathway plays a critical role in the treatment of AD. Treatment with coal tar induced epidermal cell differentiation and increased filaggrin expression and skin barrier proteins via AHR activation in primary keratinocytes from AD patients [28]. An increase in IAld, a ligand of AHR, induced by *B. longum* CCFM1029 or skin bacteria, reduced TSLP levels and further reduced Th2 cytokine levels in AD-like mice [30,43]. Furthermore, in some cohort studies, ILA was the predominant metabolite from *B. infantis* grown in human milk and regulated immune responses of human CD4+ T cells and monocytes in a dose-dependent manner by activating AHR and hydroxycarboxylic acid receptor 3 [44,45]. This suggests that ILA from microbial metabolism may affect immune function via activating the receptor signaling pathway. IPA has direct anti-inflammatory regulation effects on the immune cells via increasing IL-10 production or decreasing pro-inflammatory tumor necrosis factor expression [46]. As one of the ligands for AHR, type I interferons were produced in the central nervous system in combination with tryptophan metabolites such as IPA and IAld to activate AHR signaling in astrocytes and suppressed central nervous inflammation [47]. Although there is no direct evidence to link ILA and IPA to the development of AD, their immunomodulating effects may be the important action mechanism to alleviate AD-like symptoms. Collectively, these results suggested that *L. reuteri* DYNDL22M62 might activate AHR signaling via the increase in the expression of AHR ligands, such as ILA and IPA, to suppress aberrant Th2-type immune responses. 

Changes in tryptophan metabolism in the intestine were driven by gut microbiota, and gut microbial alterations contributed to the development of AD. In a cohort study, *Bacteroidetes*, *Xanthomonadaceae*, and *Bacteroidaceae* at the family level, and *Stenotrophomonas* and *Bacteroides* at the genus level were more abundant in infants (3–4 months old) with AD versus healthy controls [6]. However, children with eczema showed a higher Simpson’s reciprocal diversity index, a reduction of *Bacteroidetes*, and more abundant *Clostridium* clusters Ⅳ and ⅩⅣa in gut microbial alterations compared to at this age healthy children [48]. Among the mice in *Dermatophagoides farinae* extract-induced AD group, the proportion of S24-7_unclassified decreased but *Bacteroides* increased versus the control group [49]. In this study, the relative abundances of bacteria belonging to *Firmicutes* including *Lactobacillaceae*, *Erysipelotrichaceae*, and *Peptostreptococcaceae* were reduced in mice with AD but they were restored by *L. reuteri* DYNDL22M62 treatment (Figure 4C). Compared to the DNFB group, *L. reuteri* DYNDL22M62 significantly increased the proportions of the *Romboutsia* and *Ruminococcaceae* NK4A214 group, which were positively related to IPA and ILA (Figure 5B). However, *L. reuteri* DYNDL22M62 decreased the proportion of *Dubosiella*, which was negatively related to IPA but positively related to ILA (Figure 5C). This suggested that there might be a competitive relationship between these bacteria related to tryptophan metabolism. In some cohort studies, compared to the healthy controls, the proportions of *Romboutsia* and *Ruminococcaceae* were decreased in patients with allergic rhinitis and food allergy, respectively [50,51]. These results showed that *L. reuteri* DYNDL22M62 treatment regulated tryptophan metabolism-related gut microbiota that contributed to the AHR activation and further suppressed the excessive Th2-type response in mice.

There are some issues to be solved in the future. For instance, to evaluate ILA and IPA whether produced by *L. reuteri* DYNDL22M62 and/or gut microbiota, the role of gut microbiota will be evaluated using fecal microbiota transplantation at first and the role of *L. reuteri* DYNDL22M62 will be assessed using a germ-free mouse or antibiotic-treated mouse model. Furthermore, it is to be elucidated whether ILA/IPA alleviated AD dependent on AHR signaling although we have analyzed the expression of AHR in the skin lesion. In the next experiment, the effects of *L. reuteri* DYNDL22M62 will be evaluated using AHR antagonists such as CH223191 in wild-type mice or AHR-knockdown mice, and signaling molecules related to the AHR pathway need to be characterized based on different molecular biological methods. Finally, the mechanism of *L. reuteri* DYNDL22M62 to alleviate AD will be demonstrated comprehensively via gut microbial alterations and changes in immune responses.

In conclusion, *L. reuteri* DYNDL22M62 treatment regulated gut microbial composition and elevated tryptophan metabolism in the intestine to produce the AHR ligands including ILA and IPA, which activated AHR signaling to reduce aberrant Th2-type response in mice. Our findings suggest that treatment with *L. reuteri* DYNDL22M62 with the role of regulating gut bacteria and their tryptophan metabolism may be an effective alternative to alleviate AD.

## 4. Materials and Methods

### 4.1. L. reuteri Strains

*L. reuteri* strains including GDLZ105, FSDLZ12M1, FWXBH12M3, and DYNDL22M62 were received from the Culture Collection of Food Microorganisms in Jiangnan University (Wuxi, Jiangsu, China). deMan Rogosa Sharpe (MRS) broth medium was used to culture strains at 37 °C under anaerobic conditions for 16–24 h.

### 4.2. Animal Experimental Design

Specific-pathogen-free grade mice (C57bl/6, 6 weeks old, female, Charles River Laboratories, Beijing, China) were fed in a controlled facility with 12 h/12 h of light/dark cycle, the temperature of 20–26 °C, and a humidity level of 40–70%, and were free to intake a standard chow and water. Mice were randomly divided into 6 groups (n = 6) after one week of adaptation: control group, 2,4-dinitrofluorobenzene group (DNFB, Sigma-Aldrich, St. Louis, MO, USA), and four *L. reuteri* treated groups. AD symptoms were induced based on the method of our previous study [52]. Briefly, mice were treated with 0.5% DNFB solution on the dorsal skin and left ear on day 8 in all groups except for the control group (Figure 1A). Furthermore, mice were treated with 0.2% DNFB on days 12, 15, 18, and 21. Mice were treated with a control solution (acetone:olive oil = 4:1, *v*/*v*) in the control group. *L. reuteri* re-suspension solution of 0.2 mL (viable count: 1 × 10^9^ colony forming units) was fed once a day for three weeks, and mice in the control and DNFB groups were fed equal volumes of sterile saline.

### 4.3. Pathological Indicators

After sacrifice, ear thickness was measured and the skin section was stained using hematoxylin and eosin (H&E) solution after fixation in 4% paraformaldehyde. The paraffin-embedded specimens (5 μm) were stained using H&E solution after conventional alcohol dehydration and made transparent using xylene. After drying, the sections were scanned to evaluate the pathological symptoms of the skin.

### 4.4. Immune Markers

Tissue and blood samples were collected and determined to assess alterations of the immune responses. Tissue was treated using lysis buffer with protease inhibitor, after centrifuge, and the protein levels were measured using the bicinchonininc acid (BCA, Beyotime Biotechnology, Shanghai, China) kit. The results of protein were expressed as bovine serum albumin (BSA) equivalents (E). The serum samples were obtained after a 3500× rpm centrifuge for 20 min. The supernatant and serum samples were used to measure the alterations of inflammatory cytokines using the commensal ELISA kits (Sbjbio, SenBeiJia Biological Technology Co., Ltd., Nanjing, China) at 450 nm on a spectrophotometer (Thermo Fisher Scientific, Waltham, MA, USA).

### 4.5. Pretreatment and Determination for Indole Derivatives in Fecal Samples

After being vacuum freeze-dried, the fecal sample was mixed with 900 μL extracting solution (methanol:ultrapure water = 1:9, *v*/*v*) to prepare supernatant solution after homogenization treatment (65 HZ, 3 min). After vacuum concentration (45 °C, 3 h), resuspension with 200 μL extracting solution, and centrifugation (15,000× *g*, 10–15 min), the solution was treated using the 0.22 μm membrane to obtain the final sample for UHPLC Q-Exactive-MS analysis. Detailed information for the determination of indole derivatives was referred to in our previous article [43]. Briefly, a binary mobile phase consisted of acetonitrile (mobile phase A) and 0.1% formic acid (mobile phase B) in a 20 min gradient program. In positive ion mode, mass spectrometry was performed on a Q-Exactive Plus MS (Thermo Fisher Scientific, Waltham, MA, USA) operating with a full-scan acquisition from 80 to 1200 *m*/*z* with a resolution of 70,000. After determination, a quantification analysis of indole derivatives was carried out on Xcalibur 4.0 (Thermo Fisher Scientific, Waltham, MA, USA).

### 4.6. Sequencing and Analysis of Gut Microbiota

Fecal DNA was extracted using the FastDNA Spin Kit for Feces (MP Biomedicals, Santa Ana, CA, USA). It was sequenced using a high-throughput sequencing platform (Illumina, Santiago, CA, USA) based on the V3-V4 region amplification (341F and 806R). The detailed method was referred to in a previous article [53]. Briefly, fecal DNA concentration was measured using a Qubit BR dsDNA assayer (Thermo Fisher Scientific, Waltham, MA, USA). After DNA libraries preparation, they were sequenced for 500 + 7 cycles on the Illumina Miseq platform (Illumina, Santiago, California). The data were analyzed based on the quantitative insights into microbial ecology 2 (QIIME2) pipeline. The raw sequences were evaluated and screened for downstream analysis. High-quality sequences were clustered into operational taxonomic units (OTUs) and representative sequences of each cluster were used to classify and annotate bacterial taxa based on the SILVA database. PCA and LEfSe were performed to find the differences between samples and microbial biomarkers, respectively. The correlation between variations was performed by the R (free software, https://cran.rstudio.com/) package “corrplot”.

### 4.7. Statistical Analysis

Statistical analyses were processed using Statistical Product and Service Solutions version 24.0 (SPSS, IBM Corp., Armonk, NY, USA). The differences between groups were performed using one-way ANOVA and post hoc Fisher’s least significant difference (LSD) tests, * *p* < 0.05, ** *p* < 0.01, *** *p* < 0.001, and **** *p* < 0.0001. Data were presented as the mean ± SD.

## Figures and Tables

**Figure 1 ijms-23-07735-f001:**
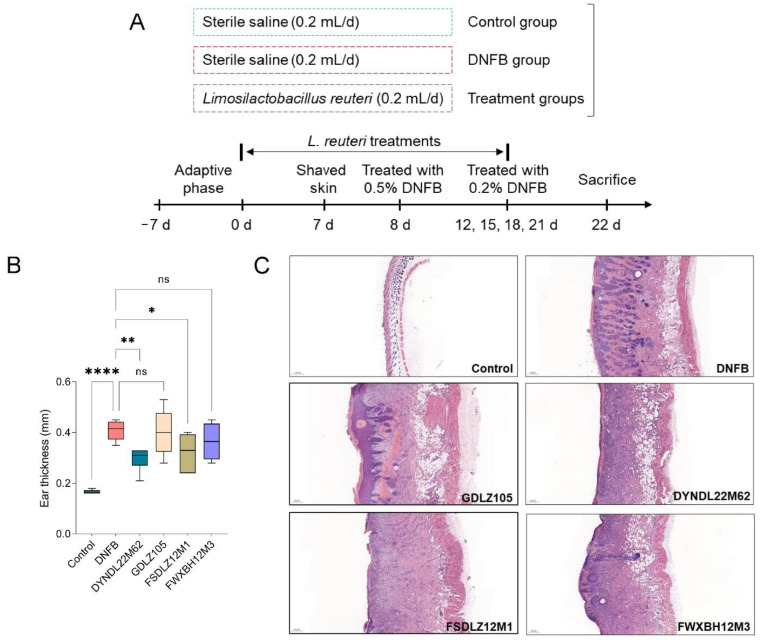
Effects of *Limosilactobacillus reuteri* on pathological symptoms of AD-like mice. (**A**) Experimental design. (**B**) Effects of *L. reuteri* strains on ear thickness of AD-like mice. (**C**) H&E staining of skin lesions, scale bar = 200 μm, original magnification = 400×. * *p* < 0.05, ** *p* < 0.01, **** *p* < 0.0001 vs. DNFB group.

**Figure 2 ijms-23-07735-f002:**
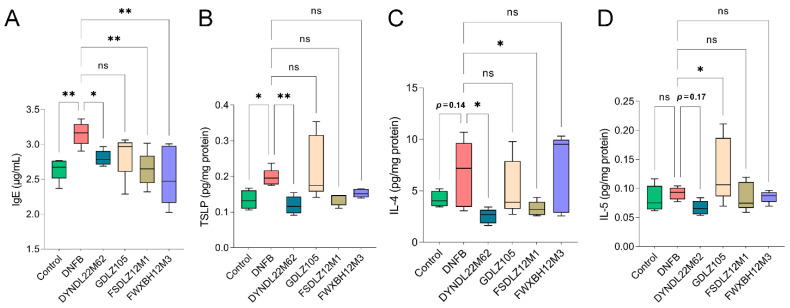
*L. reuteri* strains regulated IgE levels and the expressions of Th2 cytokines in AD-like mice. (**A**) IgE levels. (**B**–**D**) Changes in TSLP, IL-4, and IL-5 levels. * *p* < 0.05, ** *p* < 0.01 vs. DNFB group, ns, no significance. TSLP, thymic stromal lymphopoietin.

**Figure 3 ijms-23-07735-f003:**
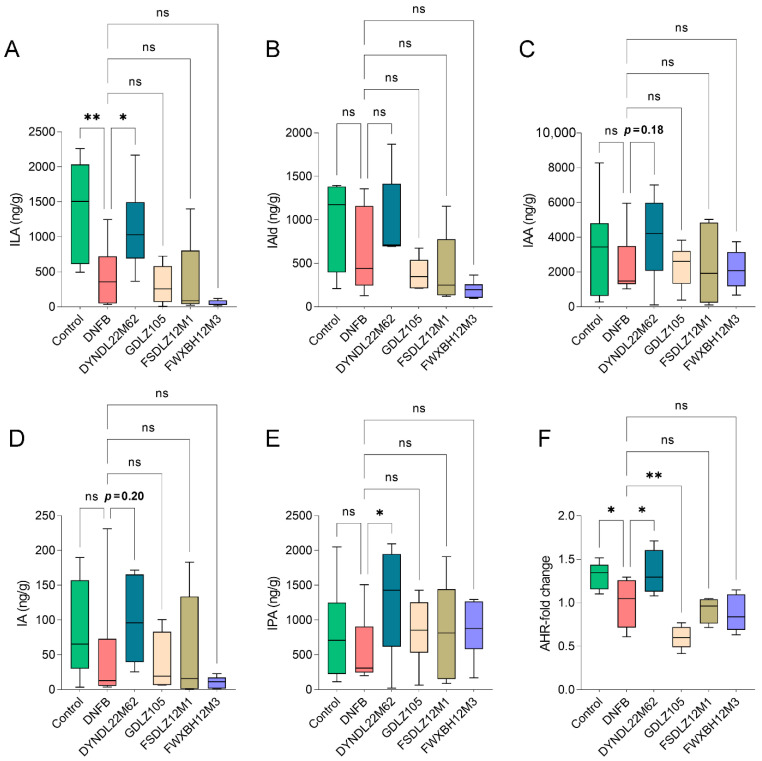
*L. reuteri* strains affected indole derivatives levels and the expression of AHR. (**A**–**E**) Changes in ILA, IAld, IAA, IA, and IPA levels in fecal samples. (**F**) Fold-change of AHR expression. * *p* < 0.05, ** *p* < 0.01 vs DNFB group, ns, no significance. ILA, indolelactic acid, IAld, indole-3-aldehyde, IAA, indoleacetic acid, IA, indoleacrylic acid, IPA, indole propionic acid.

**Figure 4 ijms-23-07735-f004:**
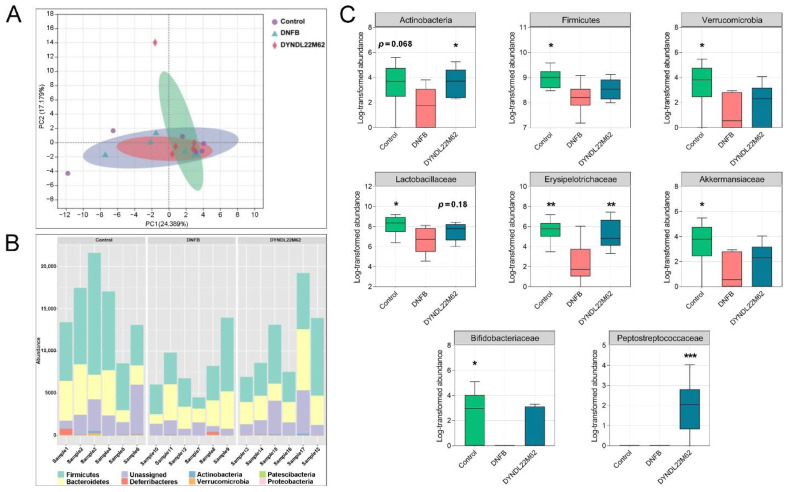
*L. reuteri* strains regulated gut microbiota in AD-like mice. (**A**) PCA plot between groups. (**B**) Gut microbial composition at the phylum level. (**C**) Effects of *L. reuteri* DYNDL22M62 on gut microbial composition. * *p* < 0.05, ** *p* < 0.01, *** *p* < 0.001 vs DNFB group. PCA, principal component analysis, PC1, principal component 1, PC2, principal component 2.

**Figure 5 ijms-23-07735-f005:**
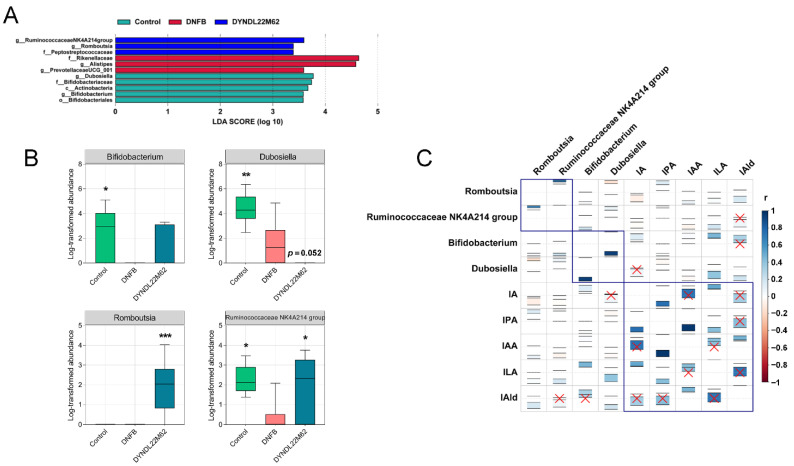
Differential gut bacteria and the correlation with indole derivatives. (**A**) Differential bacteria between groups using LEfSe analysis. (**B**) Comparison of differential bacteria. (**C**) The correlation between differential bacteria and indole derivatives (95% confidence interval, red cross, no significance). * *p* < 0.05, ** *p* < 0.01, *** *p* < 0.001 vs. DNFB group. LEfSe, linear discriminant analysis (LDA) effect size.

## Data Availability

The original contributions presented in the study are publicly available. This data can be found here: https://www.ncbi.nlm.nih.gov/bioproject/?term=PRJNA832509 (accessed on 27 April 2022), accession number: PRJNA832509.

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
