# Peer review of "Limosilactobacillus reuteri Attenuates Atopic Dermatitis via Changes in Gut Bacteria and Indole Derivatives from Tryptophan Metabolism"

_ijms, 2022, doi:10.3390/ijms23147735_

Round 1
Reviewer 1 Report
General comments:
The manuscript deals with the study of the effects of four L. reuteri strains administration on atopic dermatitis symptoms in vivo and determination of the mechanism by which these strains alleviate atopic dermatitis via the gut microbiota.
The scientific merit of the paper is appropriate and the results obtained are discussed in depth. Additionally, the description of the experiments is clear and the experimental conditions used are clearly justified. On the other hand, the data were statistically analyzed, to give an appropriate analysis of the results.
However, although the English language is correct, there are some syntax and grammatical errors throughout the manuscript that should be corrected.
For these reasons, the paper, after making minor changes, should be accepted for publication in the International Journal of Molecular Sciences.
Other considerations are as follows:
1. Title of the manuscript: Limosilactobacillus reuteri attenuate atopic dermatitis via changes in gut bacteria and indole derivatives from tryptophan metabolism.
Reviewer: Substitute attenuate by attenuates.
2. Results section.
The names of the L. reuteri strains should be rewritten in italics.
3. Discussion section.
3.1. Page 8, lines 222-223: L. reuteri strains WU and 100-23 induced intraepithelial T lymphocytes via the release of AHR ligands including ILA and reprogramed intraepithelial CD4 T cells.
Reviewer: Substitute “reprogramed” with “reprogrammed”.
3.2. Page 8, lines 255-256: Among the mice in Dermatophagoides farinae extract-induced AD group, the proportion of S24-7_uncalssified decreased but Bacteroides increased versus the control group (43).
Reviewer: Substitute “uncalssified” with “unclassified”.
3.3. Page 9, lines 264-265: This suggested that there might a competitive relationship between these bacteria related to tryptophan metabolism.
Reviewer: Insert “be” after “might”.
Author Response
1. Title of the manuscript: Limosilactobacillus reuteri attenuate atopic dermatitis via changes in gut bacteria and indole derivatives from tryptophan metabolism.
Reviewer: Substitute attenuate by attenuates.
A1: We have revised it according to your advice.
2. Results section.
The names of the L. reuteri strains should be rewritten in italics.
A2: We have checked and revised them throughout the manuscript according to your advice.
3. Discussion section.
3.1. Page 8, lines 222-223: L. reuteri strains WU and 100-23 induced intraepithelial T lymphocytes via the release of AHR ligands including ILA and reprogramed intraepithelial CD4 T cells.
Reviewer: Substitute “reprogramed” with “reprogrammed”.
A3.1: We have revised it according to your advice.
3.2. Page 8, lines 255-256: Among the mice in Dermatophagoides farinae extract-induced AD group, the proportion of S24-7_uncalssified decreased but Bacteroides increased versus the control group (43).
Reviewer: Substitute “uncalssified” with “unclassified”.
A3.2: We have revised it according to your advice.
3.3. Page 9, lines 264-265: This suggested that there might a competitive relationship between these bacteria related to tryptophan metabolism.
Reviewer: Insert “be” after “might”.
A3.3: We have inserted it in this sentence according to your advice.

Reviewer 2 Report
In such study, the Authors aimed to evaluate the effects of L. reuteri strains in Atopic Dermatitis symptoms . They found that the treatment with L. reuteri DYNDL22M62 and its tryptophan metabolism may be an effective alternative to allieviate AD.Only minor queries:
It would be interesting to linger the effects of the other gut bacteria such as Clostridium Difficile, Clostridia, S. aureus, Escherichia coli, on pathological AD symptoms and how the differences between adult gut microbiota and children gut microbiota can influence AD symptoms .
Line 33, you should add: "Susceptibility to AD seems to depend by both congenital and acquired environmental factors that promote dysfunction of the epidermal barrier and/or dysregulation of the immune response. The strongest genetic predisposing factor to AD is loss-of-function mutations in the filaggrin gene, which is essential for skin barrier function." and cite : doi: 10.1111/exd.14276 and doi: 10.1111/jdv.17964.
Author Response
In such study, the Authors aimed to evaluate the effects of L. reuteri strains in Atopic Dermatitis symptoms. They found that the treatment with L. reuteri DYNDL22M62 and its tryptophan metabolism may be an effective alternative to alleviate AD. Only minor queries:
It would be interesting to linger the effects of the other gut bacteria such as Clostridium Difficile, Clostridia, S. aureus, Escherichia coli, on pathological AD symptoms and how the differences between adult gut microbiota and children gut microbiota can influence AD symptoms.
A: We have added and discussed the effects of gut microbiota such as E. coli and Clostridium on pathological AD symptoms and the differences between adult gut microbiota and children gut microbiota in patients with AD as follows: “It has been reported that the proportion of E. coli was positively correlated to serum IgE levels (12), but Bacteroides fragilis decreased IL-4 levels produced by CD4+ T cells in germ-free mice (13). In a cohort of 24 infants, compared to healthy controls, the proportion of bacilli was significantly higher in AD infants. Furthermore, Clostridia, but not bacilli and E. coli, is significantly related to age at AD onset and negatively correlated with the proportion of eosinophils in the blood (14). However, compared with patients with AD (aged 6-22 years old), the relative abundance of Clostridium was higher but Blautia and Parabacteroides were lower in healthy controls (15). There are significantly different gut microbial taxa in infants with AD compared to that in child/adult patients.”.
Line 33, you should add: "Susceptibility to AD seems to depend by both congenital and acquired environmental factors that promote dysfunction of the epidermal barrier and/or dysregulation of the immune response. The strongest genetic predisposing factor to AD is loss-of-function mutations in the filaggrin gene, which is essential for skin barrier function." and cite : doi: 10.1111/exd.14276 and doi: 10.1111/jdv.17964
A: We have added this sentence in this paragraph according to your advice as follows: “Susceptibility to AD is closely associated with genetic and environmental factors that increase the dysfunction of the epidermal barrier and/or dysregulation of the immune response, and mutations in the filaggrin gene are the strongest genetic predisposing factor to induce dysfunction of the epidermal barrier in AD (2, 3).”.
